# Oleic Acid Copolymer as A Novel Upconversion Nanomaterial to Make Doxorubicin-Loaded Nanomicelles with Dual Responsiveness to pH and NIR

**DOI:** 10.3390/pharmaceutics12070680

**Published:** 2020-07-20

**Authors:** Jin Zhang, Xiaoyue Tang, Chuanqing Huang, Zeyu Liu, Yong Ye

**Affiliations:** Department of Pharmaceutical Engineering, School of Chemistry and Chemical Engineering, South China University of Technology, Guangzhou 510640, China; cezhangjin@mail.scut.edu.cn (J.Z.); 201920123162@mail.scut.edu.cn (X.T.); 201910104713@mail.scut.edu.cn (C.H.); 201810106054@mail.scut.edu.cn (Z.L.)

**Keywords:** oleic acid copolymer, upconversion nanomaterial, nanomicelles, drug delivery, dual responsiveness

## Abstract

Oleic acid (OA) as main component of plant oil is an important solvent but seldom used in the nanocarrier of anticancer drugs because of strong hydrophobicity and little drug release. In order to develop a new type of OA nanomaterial with dual responses to pH and near infrared light (NIR) to achieve the intelligent delivery of anticancer drugs. The novel OA copolymer (mPEG-PEI-(NBS, OA)) was synthesized by grafting OA and *o*-nitrobenzyl succinate (NBS) onto mPEGylated polyethyleneimine (mPEG-PEI) by amidation reaction. It was further conjugated with NaYF_4_:Yb^3+^/Er^3+^ nanoparticles, and encapsulated doxorubicin (DOX) through self-assembly to make upconversion nanomicelles with dual response to pH and NIR. Drug release behavior of DOX, physicochemical characteristics of the nanomicelles were evaluated, along with its cytotoxic profile, as well as the degree of cellular uptake in A549 cells. The encapsulation efficiency and drug loading capacity of DOX in the nanomicelles were 73.84% ± 0.58% and 4.62% ± 0.28%, respectively, and the encapsulated DOX was quickly released in an acidic environment exposed to irradiation at 980 nm. The blank nanomicelles exhibited low cytotoxicity and excellent biocompatibility by MTT assay against A549 cells. The DOX-loaded nanomicelles showed remarkable cytotoxicity to A549 cells under NIR, and promoted the cellular uptake of DOX into the cytoplasm and nucleus of cancer cells. OA copolymer can effectively deliver DOX to cancer cells and achieve tumor targeting through a dual response to pH and NIR.

## 1. Introduction

Oleic acid (OA) as a natural compound widely exists in plant oil such as *Camellia* seed oil, which is extracted from seeds of *Camellia oleifera* Abel and contain OA above 90% [1,2]. OA has many important physiological functions for human beings: (I) reducing blood lipid levels in patients with hyperlipidemia, cardiovascular diseases, and high cholesterol [3]. (II) Participating in the synthesis of prostaglandins which have important regulatory functions on biological organisms [4]. (III) Maintaining the physiological activity of the membrane and the normal metabolism [5]. (IV) Regulating white blood cell activity to reduce inflammation and enhance bactericidal effect [6]. In addition, OA has good biodegradability and biosafety, and can be used as solvent for drug delivery [7]. It can enhance the stability of anticancer drugs, promote the entry of anticancer drugs into cells, inhibit the metastasis of cancer and the expression of oncogenes, and increase the sensitivity of anticancer drugs to chemotherapy. It also has natural tumor-targeting properties, because OA can increase the intake of fatty acids in tumors as a source of nutrition, and has great potential in tumor treatment [5,8]. The prodrug nanoparticles based on OA modification have improved the stability of anticancer drugs and achieved the role of nutritional targeted delivery of drugs [9]. However, in clinical trials, the drug release rate of these nanoparticles at the tumor site was extremely low, and almost no free drug was released in 24 h [10,11], so that OA is seldom used in the nanocarrier of anticancer drugs. On the basis of OA pharmacological activities, it is meaningful to design OA drug carriers with controlled release at the tumor site.

On the other hand, doxorubicin (DOX), an effective chemotherapeutic drug widely used in the treatment of various cancers, can target DNA and induce cell death by preventing DNA replication and cell division [12]. However, it has serious side effects such as cardiotoxicity, mucositis, myelosuppression, and alopecia besides multidrug resistance of tumor cells [13,14]. Occurrence of the multidrug resistance is due to many factors, such as increased efflux, impaired apoptosis, decreased drug influx, and changed cell cycle regulation [15]. The development of DOX-loaded nanocarriers has become an effective strategy for improving the targeting of cancer cells, reducing the side effects of DOX, and increasing its accumulation in tumor [16]. 

Currently, various nanocarriers have been made for the delivery of anticancer drugs to enhance targetability, lipophilicity, stability, and biocompatibility, and to overcome multidrug resistance, including polymer nanomicelles [17], nanocapsules [18], nanoliposomes [19], and nanoemulsions [20]. Among them, polymer nanomicelles have become a potential anticancer drug carrier because of their outstanding drug loading, long blood circulation, enhanced permeability and retention (EPR). The polymer nanomicelles are formed by the self-assembly property of the amphiphilic copolymer in an aqueous solution, wherein the inner core is composed of a hydrophobic substance and the outer shell is composed of a hydrophilic substance [21,22]. Therefore, the hydrophobic drug can spontaneously enter the core of the nanomicelles to achieve encapsulation of the drug [23]. However, the polymer nanoparticles formed by self-assembly of the amphiphilic copolymer are unstable and easily destroyed during blood circulation, resulting in leakage of the encapsulated drug and difficulty in achieving controlled release [24]. 

The design of intelligent nanomicelles that are stable under physiological conditions but decomposed in a controlled manner under certain stimulation has attracted more and more attention [25,26]. Among these stimuli, nanomicelles with pH response are most commonly used, which utilize functional groups that are degradable under slightly acidic conditions as responsive groups to trigger drug release, such as hydrazones, esters, imines, or acetals [27,28,29]. However, the tumor targeting of nanomicelles responsive to pH is not enough in drug delivery, multiple responsiveness may improve it. Photoresponsive nanomicelles can achieve drug release in tumors by irradiation outside the body, its strategy to trigger drug release is to utilize the photolysis characteristics of the photosensitive groups. Under irradiation, the nanomicelles are unstable or even completely destroyed due to the breakage of the photosensitive groups [30]. However, most of photoresponsive materials only react with UV or visible light, the former is harmful to the human body, and the latter is too weak to penetrate the skin and reach the deeper tumor.

In this study, OA was utilized to design a drug-loaded nanomaterial of copolymer by grafting hydrophobic OA and photosensitive *o*-nitrobenzyl succinate (NBS) to hydrophilic methoxyl poly ethylene glycol (mPEG) modified polyethyleneimine (mPEGylated PEI), and conjugating with NaYF_4_:Yb^3+^/Er^3+^ nanoparticles to make upconversion nanomicelles with pH and NIR responsive properties. The results indicate that the hydrophobic anticancer drug DOX has been successfully encapsulated into the nanomicelles to achieve targeting tumor cells. The scheme is illustrated in Figure 1. The novelty of this work is design and synthesis of the OA nanomaterial with dual responsiveness to pH and NIR for the nanocarrier of anticancer drugs, which is hopeful for deep tumor targeting therapy under NIR irradiation.

## 2. Materials and Methods

### 2.1. Materials

OA was distilled from *Camellia* seed oil in our lab with its purity of 98% [31,32]. PEI with molecular weight ≈10,000, mPEG with molecular weight ≈2000, DOX, succinic anhydride, *N*-hydroxysuccinimide (NHS), 4-Nitrophenyl chloroformate, *o*-nitrobenzyl alcohol, 4-dimethylaminopyridine (DMAP), and 1-(3-dimethylaminopropyl)-3-ethylcarbodiimide hydrochloride (EDC) were purchased from Macklin Reagent Co. (Shanghai, China). Yttrium oxide (Y_2_O_3_, 99.99%), ytterbium oxide (Yb_2_O_3_, 99.99%), erbium oxide (Er_2_O_3_, 99.99%), and sodium fluoride (NaF, 98%) were purchased from Shanghai Chemical Industrial Company (Shanghai, China). Pyrene, 3-(4,5-dimethyl-2-thiazolyl)-2,5-diphenytetrazolium bromide (MTT), and 4,6-diamidino-2-phenylindole (DAPI) were purchased from Aladdin Reagent Co. (Shanghai, China). Phosphotungstic acid dye, glutaraldehyde, and copper mesh were purchased from Beijing Zhongjingkeyi Technology Co. (Beijing, China). Human lung adenocarcinoma cells A549 cells and DMEM medium were bought from Shanghai Cell Bank (Shanghai, China). Chloroform, acetonitrile, ethyl ether, and dimethyl sulfoxide (DMSO) were supplied by Sinopharm Chemical Reagent Co. (Shanghai, China). Transparent polystyrene 6-well and 96-well plates were obtained from Corning Co. (Corning, NY, USA). 

### 2.2. Synthesis of NBS

NBS was synthesized according to the previous described procedure [33]. Briefly, 4.0 g (26.12 mmol) of *o*-nitrobenzyl alcohol and 1.6 g (13.09 mmol) of *N*,*N*-dimethyl-4-aminopyridine (DMAP) were dissolved in 30 mL of dried chloroform. Then, 5.2 g (52.26 mmol) of succinic anhydride was dissolved in 10 mL of dried chloroform, and added dropwise into the previous solution. The mixture was kept stirring at 70 °C under nitrogen atmosphere for 24 h. After the completion of the reaction, the solution was evaporated to remove the excess chloroform. The concentrated chloroform solution was washed three times with hydrochloric acid (HCl) aqueous solution (10% in *v*/*v*) and then extracted with saturated sodium bicarbonate solution (NaHCO_3_). Finally, the obtained extract was acidified with 10% hydrochloric acid and 10% phosphoric acid, respectively. The white solid precipitate was collected and dried under vacuum at 40 °C for 6 h to give NBS (yield: 86%).

### 2.3. Synthesis of mPEG-Nitrophenyl Carbonate (mPEG-NO_2_)

The mPEG-NO_2_ was synthesized by mPEG reacted with 4-nitrophenyl chloroformate according to the literature with some modifications [34]. Typically, 10.0 g (5.0 mmol hydroxyl groups) of mPEG was dissolved in dried acetonitrile (60 mL), then the appropriate amount of triethylamine (2 mL) was added. A total of 2.0 g (9.9 mmol) of 4-Nitrophenyl chloroformate in acetonitrile solution (20 mL) was injected dropwise into the above solution. The mixture was reacted at 25 °C under nitrogen protection for 24 h. The reaction solution was evaporated to remove the excess acetonitrile and precipitated into excess cold ethyl ether. The product was dried under vacuum to give mPEG-NO_2_ (yield: 74%).

### 2.4. Synthesis of mPEG-PEI

The mPEG-PEI was synthesized by grafting mPEG-NO_2_ onto PEI. Briefly, 6.0 g (0.6 mmol) of PEI and 1.5 g (0.69 mmol) of mPEG-NO_2_ were dissolved in deionized water (40 mL). The mixture was stirred for 48 h at room temperature. Afterwards, the solution was dialyzed in a dialysis bag (3500 Da molecular weight cutoff) to remove unreacted reagent and small molecular species, and freeze-dried to get mPEG-PEI (yield: 85%).

### 2.5. Synthesis of mPEG-PEI-(NBS, OA)

The mPEG-PEI-(NBS, OA) was synthesized in the following procedure. First, 0.3 g (1.1 mmol) of OA and 0.2 g (0.79 mmol) of NBS were activated in the presence of EDC (0.26 g, 1.35 mmol) and NHS (0.16 g, 1.35 mmol) by stirring in DMSO (10 mL) for 2 h at room temperature. Subsequently, the solution of activated OA and NBS was dropwise added into the DMSO solution (3 mL) of mPEG-PEI (150 mg, 0.013 mmol) and stirred at room temperature for 24 h. The mixture was dialyzed in a dialysis bag (3500 Da molecular weight cutoff) for 72 h with deionized water to remove unreacted reactants, and freeze-dried to get mPEG-PEI-(NBS, OA) (yield: 85%).

### 2.6. Synthesis of mPEG-PEI-(NBS, OA) Upconversion Conjugate

The NaYF_4_:Yb^3+^/Er^3+^ UCNPs were synthesized by hydrothermal reaction following a literature protocol with slight modifications [35]. All the lanthanide nitrates were prepared by dissolving the respective rare-earth oxides in nitric acid. Typically, 1 mmol of Re (NO)_3_ (Y:Yb:Er = 78%:20%:2%), 0.84 g (20 mmol) of NaF, and 0.74 g (2 mmol) of EDTA-Na were mixed in 50 mL water, and heated to 180 °C in autoclave and kept at this temperature for 20 h. After cooling down to room temperature, the products were precipitated by ethanol, washed three times with water, and dried in vacuum. Then, 1.1 g product was collected. A total of 150 mg of mPEG-PEI-(NBS, OA) and 50 mg UCNPs were dissolved in dichloromethane, and stirred at room temperature for 3 h. mPEG-PEI-(NBS, OA) upconversion conjugate (0.2 g) was obtained by evaporation of dichloromethane and washing by ethanol.

### 2.7. Preparation of DOX-Loaded mPEG-PEI-(NBS, OA) Upconversion Nanomicelles

The DOX-loaded mPEG-PEI-(NBS, OA) upconversion nanomicelles were prepared using the membrane dialysis method. Briefly, mPEG-PEI-(NBS, OA) upconversion conjugate (45 mg) and DOX (3 mg) were dissolved together in DMSO (2 mL). Then, the mixture solution was added dropwise into 15 mL deionized water with magnetic stirring. The solution was dialyzed with deionized water for 72 h to remove the residual DMSO and unencapsulated DOX (3500 Da molecular weight cutoff). The DOX-loaded mPEG-PEI-(NBS, OA) upconversion nanomicelles were obtained and stored at 4 °C for further application. The blank nanomicelles were prepared in the above method without DOX. The encapsulation efficiency (EE%) and drug loading capacity (DLC%) of DOX in the nanomicelles were evaluated using fluorescence spectrophotometer (Hitachi F-4500, Hitachi Ltd., Tokyo, Japan). The fluorescence intensity of DOX at 560 nm was measured to quantify the DOX concentration in the solution using a pre-established calibration curve. The EE% and DLC% of DOX-loaded mPEG-PEI- (NBS, OA) nanomicelles were calculated according to the following equations:(1)$$\mathrm{EE}\;(\%)=\frac{\mathrm{Weight}\;\mathrm{of}\;\mathrm{encapsulated}\;\mathrm{DOX}}{\mathrm{Weight}\;\mathrm{of}\;\mathrm{total}\;\mathrm{DOX}}\times 100\%$$
(2)$$\mathrm{DLC}\;(\%)=\frac{\mathrm{Weight}\;\mathrm{of}\;\mathrm{encapsulated}\;\mathrm{DOX}}{\mathrm{Weight}\;\mathrm{of}\;\mathrm{total}\;\mathrm{DOX}\;\mathrm{and}\;\mathrm{vector}}\times 100\%$$

### 2.8. Structural Analysis and Morphological Observation

The ^1^H NMR spectra of the synthesized NBS, mPEG-PEI and mPEG-PEI-(NBS, OA) were detected using Bruker-600 NMR spectrometer (Bruker 600 MHz, Bruker Ltd., Karlsruhe, Germany). All the samples (NBS, mPEG-PEI and mPEG-PEI-(NBS, OA)) were dissolved in deuterated DMSO (DMSO-d6) and determined by the NMR spectrometer. The FTIR spectra of NBS, mPEG-PEI, and mPEG-PEI-(NBS, OA) were determined by FTIR spectrometer (Bruker VERTEX70, Bruker Ltd., Karlsruhe, Germany) in the range of 4000–400 cm^−1^. 

The particle sizes and zeta potential of the blank, DOX-loaded, weakly acidic pH, and NIR stimulated mPEG-PEI-(NBS, OA) upconversion nanomicelles were measured by Malvern Zetasizer Nano ZS90 (Malvern Instruments Ltd., Malvern, UK). The suitable samples were dispersed in deionized water and then measured at 25 °C to show a refractive index 1.33. Data were calculated as the average of three repetitions. Morphological properties of the nanomicelles were observed by transmission electron microscopy (TEM) (JEM-1400 Plus, JEOL, Tokyo, Japan). Before observation, one droplet of the sample suspension was deposited onto a copper grid and dried at room temperature, following negative staining with one drop of 2% aqueous solution of sodium phosphotungstate for contrast enhancement.

### 2.9. Determination of Critical Micelle Concentration

The critical micelle concentration (CMC) of mPEG-PEI-(NBS, OA) upconversion nanomicelles was determined by a fluorescence spectrophotometer (Hitachi F-4500, Hitachi Ltd., Tokyo, Japan). Pyrene was used as the fluorescent probe [36]. Typically, the mPEG-PEI-(NBS, OA) upconversion nanomicelles (1 mg/mL) were dispersed in deionized water to prepare the aqueous solutions with various concentration of the nanomicelles (from 0.05 to 0.25 mg/mL). Then, 100 μL of pyrene acetone solution (6 × 10^−5^ M) was added into 10 mL of above nanomicelles solution and evaporated overnight to remove the acetone. The fluorescence intensity of pyrene in different concentrations of the nanomicelles solution was determined using a fluorescence spectrophotometer. The excitation wavelength was set to 339 nm, and the fluorescence intensity was detected at 372 and 383 nm. The fluorescent intensity ratio of the first vibronic peak (372 nm) to the third vibronic peak (383 nm) of pyrene was used as an index to determine the critical micelle concentration of the nanomicelles.

### 2.10. Dual-Responsiveness Test

The pH and photo responsiveness of the mPEG-PEI-(NBS, OA) upconversion nanomicelles were evaluated by dynamic light scattering (DLS) method. Briefly, 2 mL of the nanomicelles solution purified through 0.45 μm filter was made up to 10 mL with PBS solutions of pH 7.4 and 5.5, respectively. The above solution was continuously irradiated at 980 nm (100 mW/cm^2^) for 10 min and incubated at 37 °C for 2 h. As a control, the samples without irradiation were also prepared. The particle size change of the nanomicelles was monitored using a particle size analyzer.

### 2.11. Dual-Responsive Drug Release Test

The release efficiency of DOX from DOX-loaded mPEG-PEI-(NBS, OA) upconversion nanomicelles was determined by dynamic dialysis method. The nanomicelles (1 mL) were transferred to a dialysis bag (3500 Da molecular weight cutoff) and irradiated at 980 nm (100 mW/cm^2^) for 10 min, and then dialyzed with PBS solution (50 mL) of pH 7.4 and 5.5, and incubated at 37 °C with shaking at speed of 100 rpm. At the predetermined time interval, the dialysis buffer solution (1 mL) was taken out and supplemented corresponding fresh buffer to restore original volume. The obtained samples solution was diluted to 5 mL with DMSO, and then the amount of released DOX was determined by fluorescence spectrophotometer. Meanwhile, the DOX release efficiency from the nanomicelles without radiation under the above two pH conditions was also determined in the same method.

### 2.12. Cell Culture and Cytotoxicity Assay

The cytotoxicity of mPEG-PEI-(NBS, OA) nanomicelles was estimated by MTT viability assay against A549 cells [37]. Briefly, the A549 cells were seeded in a 96-well plate at a density of 1 × 10^4^ cells per well and incubated in DMEM medium at 37 °C in the presence of 5% CO_2_ for 24 h. After removing the culture medium, the cells were treated with 100 μL of fresh medium containing different concentrations of the nanomicelles (the corresponding concentration was 0.5, 1, 2, 4, 8, 16, 32, 64, 128, and 256 μg/mL) for 24 h. Subsequently, the medium containing the nanomicelles was removed, and the cells were washed three times with PBS solution. Afterwards, the cells were added with 100 µL fresh medium and 20 μL of MTT solution (5 mg/mL), and incubated at 37 °C for 4 h to form formazan crystals. Finally, the culture medium was replaced with DMSO to dissolve the formazan crystals. The absorbance was measured at 490 nm using an enzyme-linked immunosorbent assay reader (Multiskan Cytation5, BioTek Instruments, Inc., Winooski, VT, US), and untreated cells were used as the negative control. Similarly, the cytotoxicity of free DOX and DOX-loaded mPEG-PEI-(NBS, OA) upconversion nanomicelles with or without irradiation at DOX concentration of 0.5, 1, 2, 4, 6, 8, and 10 µg/mL on A549 cells were determined using the same method. The cell viability was quantitatively calculated using the following formula:(3)$$\mathrm{Cell}\;\mathrm{viability}\;(\%)=\frac{\mathrm{Absorbance}\;\mathrm{of}\;\mathrm{test}\;\mathrm{group}}{\mathrm{Absorbance}\;\mathrm{of}\;\mathrm{control}\;\mathrm{group}}\times 100\%$$

### 2.13. Cellular Uptake Test 

In order to evaluate the cellular uptake of DOX-loaded mPEG-PEI-(NBS, OA) upconversion nanomicelles, the A549 cells were seeded in 6-well plates at a density of 1 × 10^4^ cells/well and incubated in DMEM medium with 10% fetal bovine serum (FBS) at 37 °C in the presence of 5% CO_2_ for 24 h. Subsequently, the culture media was replaced with fresh medium containing free DOX and DOX-loaded mPEG-PEI-(NBS, OA) upconversion nanomicelles (DOX concentration: 10 µg/mL) and incubated at 37 °C for 4 h, respectively. After cellular uptake, the A549 cells were washed thrice with PBS buffer to remove the extracellular free DOX or DOX-loaded mPEG-PEI-(NBS, OA) upconversion nanomicelles and fixed with fresh 2.5% glutaraldehyde for 30 min. Then, the A549 cells were stained with 4′,6-diamidino-2-phenylindole (DAPI) for 15 min. The DAPI fluorescence (blue) and DOX fluorescence (red) inside the A549 cells were observed by an inverted fluorescence microscope (Japan Olympus Co., Ltd., Tokyo, Japan) to localize DOX.

### 2.14. Statistical Analysis

All data were expressed as mean ± standard deviation, and analyzed with SPSS17.0 software (IBM Corp., Amonk, NY, USA) to obtain statistical results. The significant levels of EE% and DLC%, average size and zeta potential, drug release percentage, and cell survival rate were calculated using one-way ANOVA and Student–Newman–Keuls (SNK) tests.

## 3. Results and Discussion

### 3.1. Characterization of OA Copolymer

The mPEG-PEI-(NBS, OA) was synthesized following the strategies depicted in Figure 2. First, the succinic anhydride was reacted with the hydroxyl group of *o*-nitrobenzyl alcohol under the catalysis of DMAP to obtain NBS. The chemical structure of NBS is characterized by FTIR and ^1^H NMR. The new peaks at 1738 and 1709 cm^−1^ in the FTIR spectra demonstrated the formation of ester bonds, indicating the existence of ester groups in NBS. In addition, NBS exhibits a different –OH peak from *o*-nitrobenzyl alcohol at around 3053 cm^−1^, which is attributed to the introduction of a carboxyl group in NBS (Figure 3). Correspondingly, the ^1^H NMR spectra of NBS in Figure 4 shows the signal peaks of aromatic protons, benzyl methylene protons, and methylene protons adjacent to the carboxylic group appeared at δ 7.47–8.12, δ 5.56, and δ 2.74 ppm, respectively. The above results indicate that succinic anhydride is successfully grafted onto the hydroxyl group of *o*-nitrobenzyl alcohol to form NBS.

Thereafter, mPEG-NO_2_ was synthesized by the terminal hydroxyl groups of mPEG reacting with 4-nitrophenyl chloroformate in acetonitrile. The mPEG-NO_2_ was further coupled with the primary amine groups in PEI to form mPEG-PEI (Figure 2). Figure 3 shows the FTIR spectra of mPEG, PEI, and mPEG-PEI, respectively. The absorption peak at around 1660 cm^−1^ in the FTIR spectra of mPEG-PEI was apparently due to the absorption of the amide group. The absorption peak at 3604 cm^−1^ corresponds to the stretching vibration of –NH. There is a significant difference between the absorption peaks of mPEG-PEI with mPEG and PEI. Figure 4 illustrates the ^1^H NMR spectra of PEI and mPEG-PEI, respectively. Compared with PEI, a new proton peak appeared in the ^1^H NMR spectra of mPEG-PEI, which belongs to the unit proton peak of –O–CH_2_–CH_2_– in mPEG at δ 3.41–3.65 ppm. These results indicate that mPEG has been successfully grafted onto the PEI surface. 

Finally, mPEG-PEI- (NBS, OA) was synthesized by the remaining primary amines of mPEG-PEI reacting with the carboxyl groups of OA and NBS in the presence of EDC and NHS. The carboxyl groups of OA and NBS were first activated by EDC and then coupled with the primary amino groups in mPEG to form an amide linkage. As shown in Figure 3, the mPEG-PEI-(NBS, OA) had a broad absorption peak at around 3440 cm^−1^, which confirms the possible presence of –NH, Ar–H, –C=C–H– groups in the compound. The sharp peak at 1741 and 1641 cm^−1^ could be assigned as an amide bond (–CO–NH–), indicating that the OA and NBS are coupled to the mPEG-PEI surface in the form of an amide bond. The absorption peak at 1571–1401 cm^−1^ corresponds to the stretching vibration of –C=C– on the aromatic ring. Compared with NBS and mPEG-PEI, mPEG-PEI-(NBS, OA) showed abundant characteristic absorption peaks. The mPEG-PEI-(NBS, OA) displayed typically absorption peaks derived from both OA and NBS. The ^1^H NMR spectra of mPEG-PEI-(NBS, OA) in Figure 4 clearly showed the signals of aromatic protons, benzyl methylene protons, succinic acid methylene protons, polyethylene glycol methylene protons, and OA olefin protons. The above results confirm that the target molecules were successfully synthesized.

The mPEG-PEI-(NBS, OA) upconversion conjugate was made from the binding of mPEG-PEI-(NBS, OA) with NaYF4:Yb^3+^/Er^3+^ nanoparticles, which have been characterized in our previous research [38]. NBS has a carboxyl group and nitro group, and easily coordinate with Yb^3+^/Er^3+^ to form a stable conjugate. 

### 3.2. Characterization of the Upconversion Nanomicelles

The upconversion nanomicelles with dual response to pH and NIR were formed by self-assembly of the mPEG-PEI-(NBS, OA) conjugate. The blank nanomicelles showed an average particle size of 34.85 ± 3.23 nm with a unimodal distribution (PDI = 0.20 ± 0.03) and its zeta potential was 29.88 ± 1.54 mV. The DOX-loaded nanomicelles exhibited relatively larger hydrodynamic diameter (58.56 ± 0.54 nm) than blank nanomicelles, because the hydrophobic DOX was inserted into the core of the nanomicelles, and its zeta potential became 23.42 ± 1.25 mV. The particle size of DOX-loaded nanomicelles was in the reasonable size range for accumulating readily in tumor vasculature via the enhanced permeation and retention (EPR) effect [39]. In addition, the average particle size of nanomicelles stimulated by weakly acidic pH and NIR is about 1 μm, which indicates that weakly acidic pH and NIR can promote the disintegration of nanomicelles.

The morphology of blank, DOX-loaded, pH, and NIR stimulated nanomicelles observed by TEM (Figure 5). The results showed that the blank and DOX-loaded nanomicelles had a nearly spherical shape, and the pH and NIR stimulated nanomicelles expanded and cracked, and the size seemed to agree with that measured by DLS. All the above results indicate the good formation of the nanomicelles and demonstrate their potential application in drug delivery systems.

### 3.3. Critical Micelle Concentration (CMC) of mPEG-PEI-(NBS, OA) Upconversion Conjugates

Pyrene is a hydrophobic fluorescent substance, its ratio of the fluorescence intensity of the first and third electronic vibrational peaks (I_1_/I_3_) is strongly dependent on the polarity of the environment in which it is placed. Therefore, the CMC of the copolymer can be determined by a mutation of the ratio of the fluorescence intensity. As shown in Figure 6, the fluorescence intensity ratio of pyrene was mutated with the increase of mPEG-PEI-(NBS, OA) concentration, indicating the formation of nanomicelles and encapsulation of pyrene in the hydrophobic core. The CMC value was estimated from the threshold concentration of self-assembled nanomicelles. The results showed that the CMC of the mPEG-PEI-(NBS, OA) conjugate in water was about 0.0123 mg/mL. Below this concentration point, the mPEG-PEI-(NBS, OA) upconversion conjugate was present in the form of a monomer. Above this concentration point, the amphiphilic mPEG-PEI-(NBS, OA) conjugate began to form the nanomicelles spontaneously in aqueous solution. 

### 3.4. Dual-Responsiveness of mPEG-PEI- (NBS, OA) Upconversion Nanomicelles

Protonation is a kind of drug release mechanism in pH-sensitive drug delivery systems. The pH sensitivity of polymer nanomicelles is achieved by introducing protonated groups on the polymer chain such as amine group, imidazole group, sulfonic acid group, and carboxyl group [40,41]. When such polymer nanomicelles are in a slightly acidic environment of the tumor, the polymer will accept protons to destabilize the nanomicelle structure, thereby causing precipitation, aggregation, or depolymerization, and rapid release of the entrapped drug [42,43]. Significantly, the hydrophilic cationic polymer PEI is an excellent carrier material for non-viral genes and anticancer drugs [44]. In the slightly acidic environment of the tumor, the tertiary amino group on the PEI fragment will be protonated, and increases the charge density on the surface of nanomicelles. Electrostatic repulsion between the PEI fragments will cause the hydrophilic shell of nanomicelles to swell and loosen the hydrophobic core, resulting in a significant increase in drug release rate [45,46,47]. Compared with negatively charged carrier materials, positively charged PEI is more susceptible to be internalized by cancer cells due to its high affinity with negatively charged cell membranes and facilitates drug-loaded nanomicelles into the cancer cells for better release of anticancer drugs with nucleic acid toxicity. In addition, PEI can mask some of the positive charge after methoxyl poly ethylene glycol (mPEG) modification, which can greatly reduce cytotoxicity. The mPEG approved by the FDA for human use has no toxicity, good hydrophilicity, biodegradability, and biocompatibility. Nanomicelles coated with mPEG were often used as delivery vehicle for hydrophobic drugs, which can avoid recognition by the endothelial reticulocyte system, prolong blood circulation time, and reduce liver accumulation [48]. Irradiation at 365 nm has a strong stimulating effect on the nanomicelles containing *o*-nitrobenzyl succinate (NBS), because it can trigger the photolysis reaction of the nitrobenzyl-containing polymer, thereby destroying the integrity of the nanomicelles [49]. *o*-nitrobenzyl group in NBS become unstable nitroso group through intramolecular changes under UV irradiation, eventually leading to group cracking. When NBS is conjugated with NaYF_4_:Yb^3+^/Er^3+^ upconversion nanoparticles, NIR (980 nm) can upconvert to 365 nm light and excite NBS [38]. Therefore, OA is designed to combine with PEI, mPEG, and NBS to synthesize a new nanomaterial of mPEG-PEI-(NBS, OA), and then conjugate with NaYF_4_:Yb^3+^/Er^3+^ upconversion nanoparticles so as to keep the nanomicelles responsive to pH and NIR. The UV-responsiveness of NBS is a self-immolative process, and the mPEG-PEI-(NBS, OA) upconversion conjugate is a kind of self-immolative polymer, which can be used as biolabeling, drug delivery, and prodrugs, etc. [50].

The experimental results showed that the average particle size of the mPEG-PEI-(NBS, OA) upconversion nanomicelles was about 34.85 ± 3.23 nm at pH 7.4. When 980 nm irradiation was applied simultaneously at pH 7.4, the average particle size of the nanomicelles increased to 85.63 ± 3.78 nm, attributing to the cleavage of the photosensitive side chain NBS groups inside the nanomicelles core. This change weakens the hydrophobic force inside the nanomicelles, but retains the original core-shell structure. When the pH was lowered to 5.5, the tertiary amino group on PEI was protonated due to acidic conditions, causing the particle size of the nanomicelles suddenly increase to 843.56 ± 15.28 nm. This is because the charge density of the surface of the nanomicelles increases after PEI is protonated, and the outer shell of the nanomicelles expands under the action of electrostatic repulsion, causing flocculation suspension of the nanomicelles. pH 5.5 plus irradiation increased the average particle size of the nanomicelles up to 1023.41 ± 12.23 nm. Obviously, irradiation at the same pH produces a dual effect of protonation of PEI and cleavage of NBS side groups, resulting in a further increase in particle size and more flocculent agglomeration. These results indicate that mPEG-PEI-(NBS, OA) nanomicelles have excellent pH and photo responsiveness, and irradiation and weakly acidic conditions significantly affect the integrity of nanomicelles.

### 3.5. Drug Loading of mPEG-PEI-(NBS, OA) Upconversion Nanomicelles

DOX was loaded into the core of the mPEG-PEI- (NBS, OA) upconversion nanomicelles because of hydrophobic interaction between the hydrophobic core of the nanomicelles and the drug molecule. The encapsulation efficiency (EE%) and drug loading capacity (DLC%) of the DOX-loaded mPEG-PEI-(NBS, OA) upconversion nanomicelles were determined by fluorescence spectroscopy. The fluorescence emission spectra of different concentrations of DOX are shown in Figure 7, the standard curve was obtained from the fluorescence intensity at emission peak (560 nm) vs. DOX concentration, and a good linear relationship was obtained to calculate the encapsulation efficiency and drug loading. The EE% and DLC% of the DOX-loaded nanomicelles were 73.84% ± 0.58% and 4.62% ± 0.28%, respectively. The results indicate that DOX is successfully loaded and the mPEG-PEI-(NBS, OA) upconversion nanomicelles can be used as nano delivery vehicle for DOX in drug delivery systems.

### 3.6. Drug Release of DOX-Loaded mPEG-PEI-(NBS, OA) Upconversion Nanomicelles

The release of DOX from DOX-loaded mPEG-PEI-(NBS, OA) upconversion nanomicelles was investigated under NIR in simulated tumor microenvironment. Figure 8 shows that different pH (7.4 and 5.5) and irradiation (980 nm) triggered drug release from DOX-loaded mPEG-PEI-(NBS, OA) nanomicelles. At pH 7.4 without irradiation, only 10.85% of DOX was released within 24 h from the nanomicelles, indicating that DOX is steadily encapsulated within nanomicelles under simulated physiological conditions. However, under pH 7.4 and irradiation, the cumulative release rate of DOX in the same time was approximately 30.43%, indicating that photolysis of the NBS side groups under irradiation causes the weakening of the hydrophobic force inside the DOX-loaded nanomicelles core to trigger the release of DOX. Similarly, at pH 5.5 without irradiation, the protonation of PEI in the nanomicelles destroyed the nanomicelles structure, and the cumulative release rate of the drug reached 80.86% at 24 h. Obviously, when pH 5.5 and irradiation acted together on the DOX-loaded nanomicelles, the burst release behavior of DOX was observed due to rapid depolymerization of the nanomicelles. The total amount of DOX released after 24 h reached 90.43%, which is due to acid-stimulated PEI protonation and photolysis of NBS. Overall, the most rapid release was found when the dual stimuli were applied simultaneously.

### 3.7. Cytotoxicity of mPEG-PEI-(NBS, OA) Upconversion Nanomicelles

The cytotoxicity of blank and DOX-loaded mPEG-PEI-(NBS, OA) upconversion nanomicelles was evaluated by MTT assays on A549 cells. No obvious cytotoxicity was observed in Figure 9 after A549 cells were incubated with different concentrations of blank mPEG-PEI-(NBS, OA) upconversion nanomicelles for 24 h with or without irradiation at 980 nm. When the concentration of nanomicelles increased to 256 μg/mL, the cell viability remained above 95.97%. The result indicates that the blank mPEG-PEI- (NBS, OA) upconversion nanomicelles have safety and excellent biocompatibility, and can be used as nontoxic nanocarriers. The upconversion nanomicelles are safe because they are made from OA, mPEG, and NaYF_4_:Yb^3+^/Er^3+^ nanoparticles, which have biocompatibility and are easily removed from the organism [51]. On the other hand, free DOX and DOX-loaded mPEG-PEI-(NBS, OA) upconversion nanomicelles exhibited a dose-dependent increase of cytotoxicity on the A549 cells (Figure 10). The DOX-loaded mPEG-PEI-(NBS, OA) upconversion nanomicelles can achieve enhanced cell killing effects compared to free DOX. It is worth noting that the cell viability of A549 cells treated with DOX-loaded mPEG-PEI-(NBS, OA) upconversion nanomicelles under irradiation was only 23.58%, which was significantly lower than that without irradiation. In addition, there was little difference in the cytotoxicity of free DOX on A549 cells in the presence or absence of irradiation. The results indicate that DOX-loaded mPEG-PEI- (NBS, OA) upconversion nanomicelles can increase cytotoxicity after irradiation. According to the literature, the extracellular environment of the tumor is acidic (pH 6.5), and the pH in the endosomes and lysosomes is even lower (pH 5.0–5.5). Therefore, the nanomicelles structure is destroyed after the DOX-loaded mPEG-PEI-(NBS, OA) upconversion nanomicelles is internalized by endocytosis of cancer cells. Combined with irradiation, the nanomicelles can completely decompose and release drugs, thereby achieving higher anticancer activity. 

### 3.8. Cellular Uptake of DOX-Loaded mPEG-PEI-(NBS, OA) Upconversion Nanomicelles

The accumulation of DOX and DOX-loaded mPEG-PEI-(NBS, OA) upconversion nanomicelles in human lung epithelial carcinoma A549 cells was investigate by fluorescence microscopy. The cell nuclei were stained with DAPI (blue) and the intrinsic fluorescence intensity (red) of DOX was used to monitor qualitatively accumulation and release in the cells. As shown in Figure 11, significant DOX fluorescence was observed in the cytoplasm of A549 cells after 4 h of incubation with free DOX, but no free DOX was observed in the nucleus. The above results revealed that free DOX can easily enter the cytoplasm of A549 cells, but it is difficult to enter the nucleus. Compared to free DOX, DOX-loaded mPEG-PEI-(NBS, OA) upconversion nanomicelles significantly increased intracellular accumulation of DOX in A549 cells, and obvious DOX fluorescence was observed in both cytoplasm and nucleus of A549 cells (Figure 11). It is well known that DOX is a DNA toxin that can target nuclear DNA to cause its damage or inhibit topoisomerase-induced cell death [52]. Therefore, DOX can be efficiently delivered to the cytoplasm and nucleus of tumor cells after being encapsulated by the nanomicelles to produce higher cytotoxicity against tumor cells. These results indicate that DOX-loaded mPEG-PEI-(NBS, OA) upconversion nanomicelles could be effectively internalized by A549 cells through passive targeting.

## 4. Conclusions

The OA copolymer mPEG-PEI-(NBS, OA) conjugate was successfully synthesized by grafting hydrophobic OA onto hydrophilic mPEG-PEI and then interfacing with photosensitive NBS and NaYF_4_:Yb^3+^/Er^3+^ nanoparticles. Then, the novel drug-loaded upconversion nanomicelles were made by the self-assembly of mPEG-PEI-(NBS, OA) conjugate in aqueous solution for the delivery of DOX. The properties of mPEG-PEI-(NBS, OA) upconversion nanomicelles characterized by size distribution, zeta potential, CMC, EE%, DLC%, and TEM confirm the good formation of nanomicelles and potential application in drug delivery systems. The rapid release of DOX shows dual responsiveness of the nanomicelles in the acidic environment with irradiation at 980 nm. The MTT assay of A549 cells confirms that the mPEG-PEI-(NBS, OA) upconversion nanomicelles have safety and excellent biocompatibility, and can be considered as non-toxic nanocarriers. However, the DOX-loaded mPEG-PEI-(NBS, OA) upconversion nanomicelles increase cytotoxicity under irradiation at 980 nm. The positively charged nanomicelles have a high affinity with negatively charged cell membranes, and can be internalized by cancer cells due to breakage of the nanomicelles and release of OA under irradiation. Due to OA’s lipophilicity and natural targeting, mPEG-PEI-(NBS, OA) upconversion nanomicelles can transport DOX to tumor cells and release under dual stimulation of pH and NIR, and produce high cytotoxicity to cancer cells. This new nanomaterial of OA copolymer can be used for anticancer drug delivery.

## Figures and Tables

**Figure 1 pharmaceutics-12-00680-f001:**
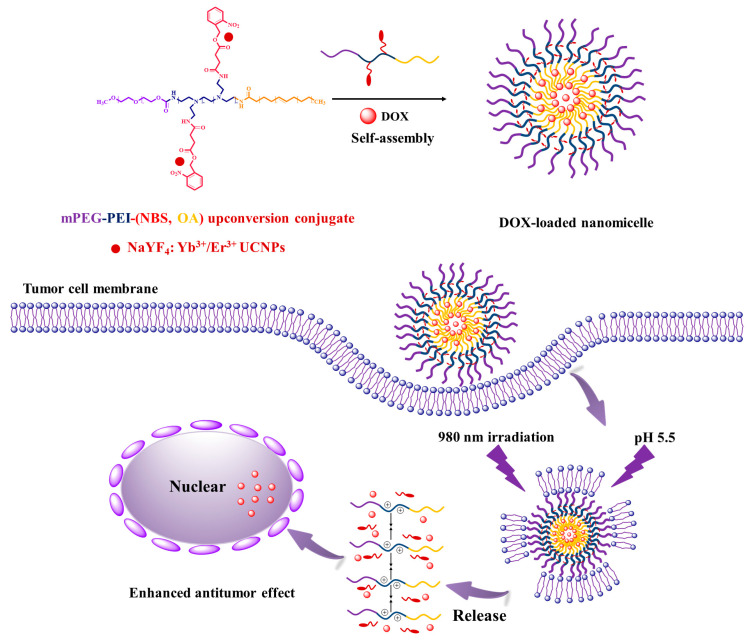
Scheme of doxorubicin (DOX)-loaded oleic acid (OA) copolymer upconversion nanomicelles penetrating the cell membranes and delivering DOX into tumor cells.

**Figure 2 pharmaceutics-12-00680-f002:**
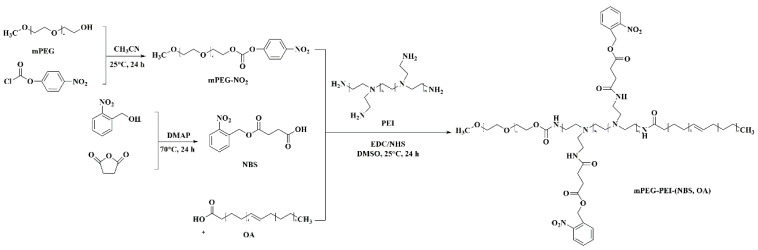
Synthetic route of OA copolymer mPEG-PEI-(NBS, OA).

**Figure 3 pharmaceutics-12-00680-f003:**
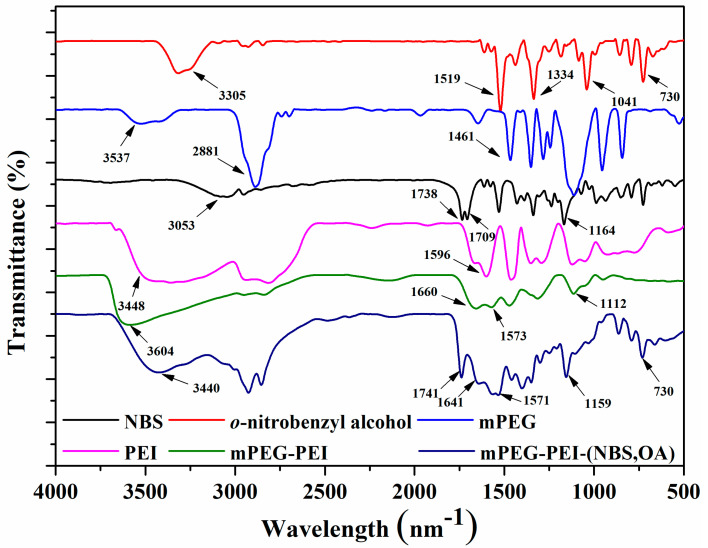
FTIR spectra of *o*-nitrobenzyl alcohol, NBS, mPEG, PEI, mPEG-PEI, and mPEG-PEI-(NBS, OA).

**Figure 4 pharmaceutics-12-00680-f004:**
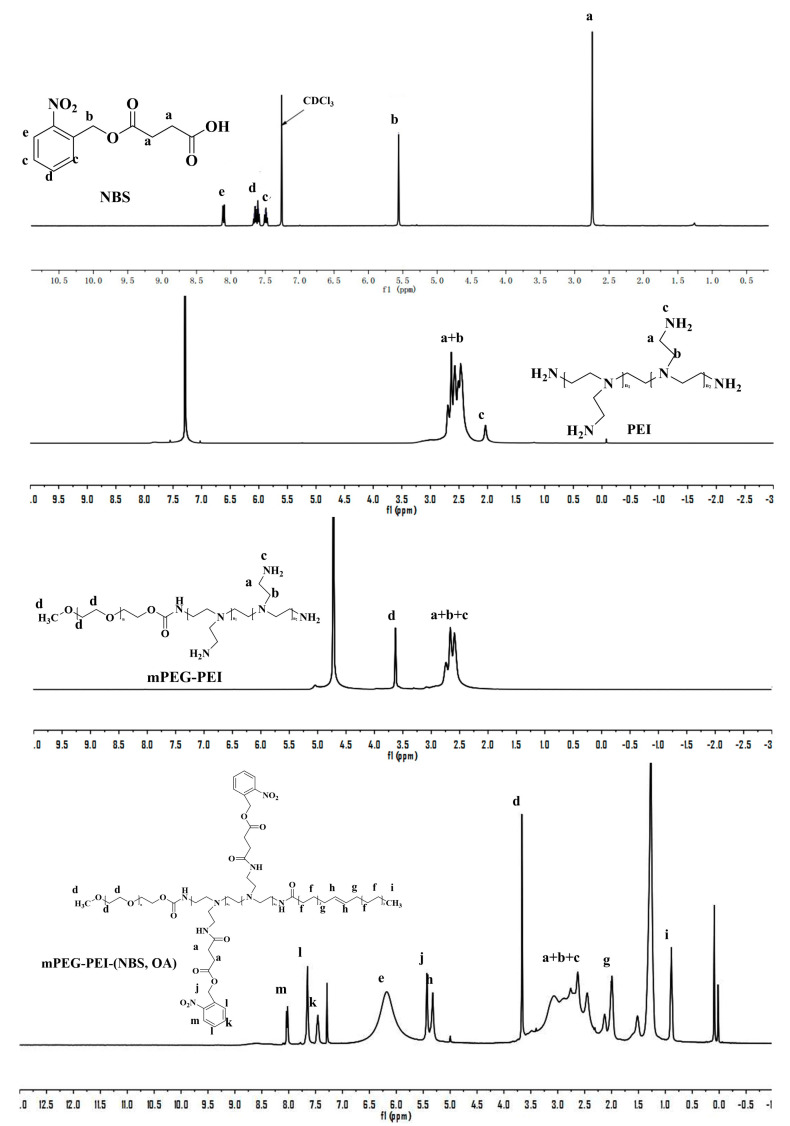
^1^H NMR of *o*-nitrobenzyl succinate (NBS), PEI, mPEG-PEI, mPEG-PEI-(NBS, OA).

**Figure 5 pharmaceutics-12-00680-f005:**
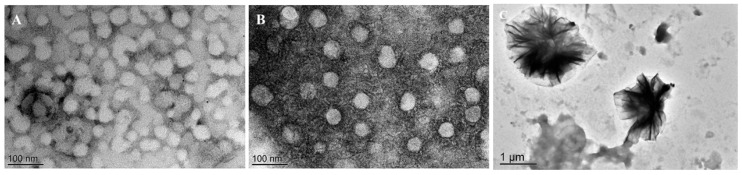
Transmission electron microscopy (TEM) images of blank (**A**), DOX-loaded (**B**), pH and near infrared light stimulated (**C**) mPEG-PEI-(NBS, OA) upconversion nanomicelles.

**Figure 6 pharmaceutics-12-00680-f006:**
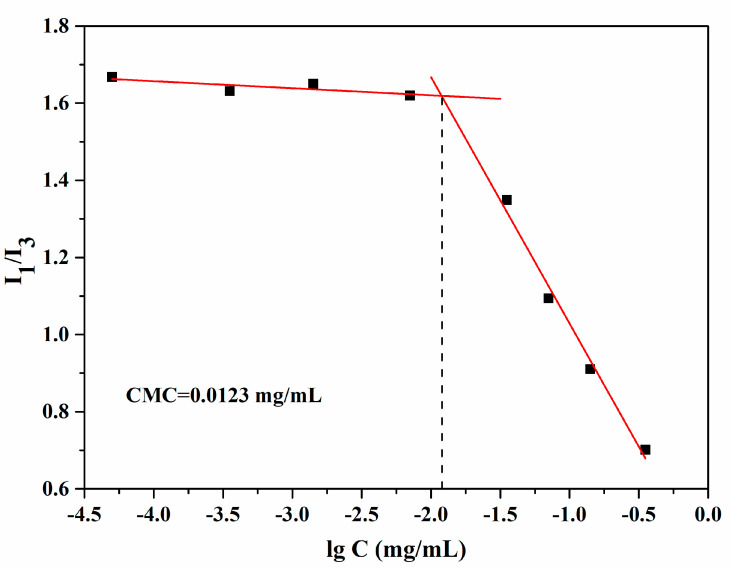
Critical micelle concentration of blank mPEG-PEI-(NBS, OA) upconversion nanomicelles.

**Figure 7 pharmaceutics-12-00680-f007:**
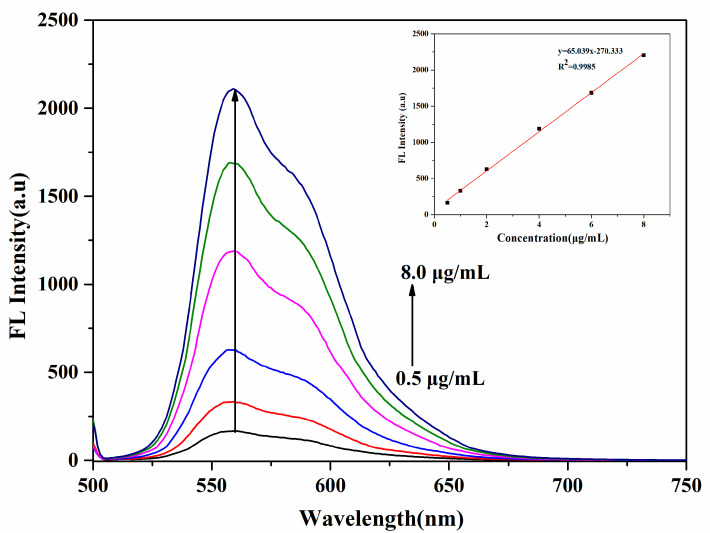
Fluorescence spectra of different concentrations of DOX and its standard curve.

**Figure 8 pharmaceutics-12-00680-f008:**
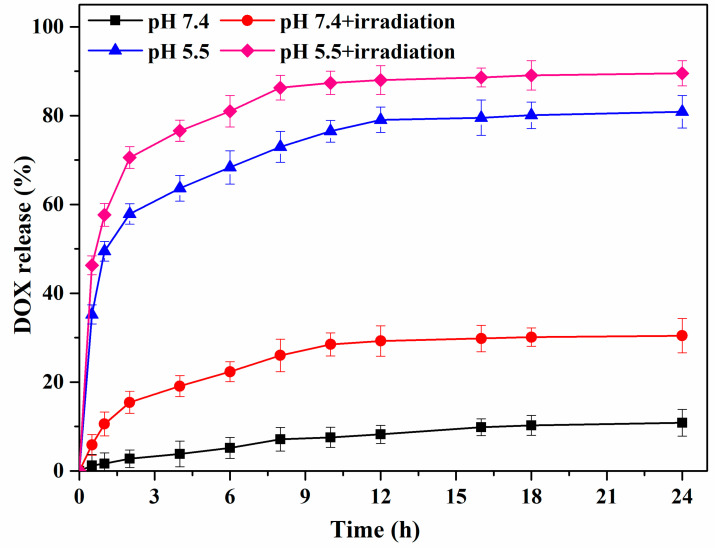
DOX release curves from the mPEG-PEI-(NBS, OA) upconversion nanomicelles with or without irradiation at 980 nm.

**Figure 9 pharmaceutics-12-00680-f009:**
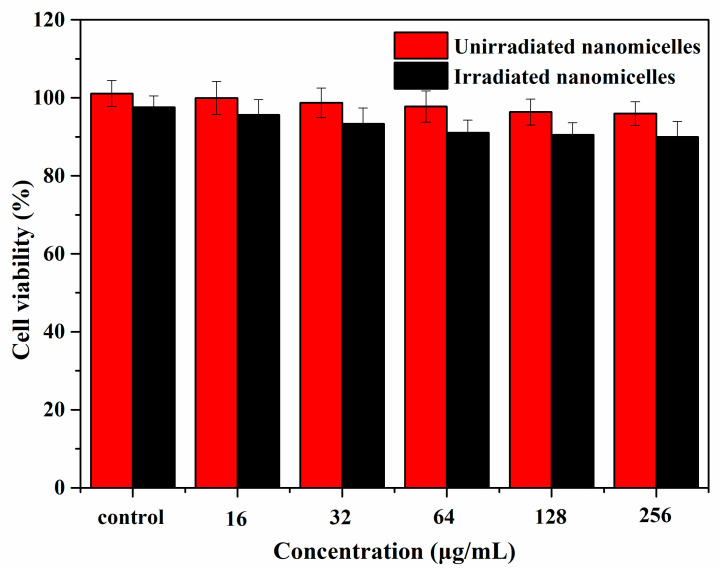
Cytotoxicity of blank mPEG-PEI-(NBS, OA) upconversion nanomicelles with or without irradiation at 980 nm against A549 cells.

**Figure 10 pharmaceutics-12-00680-f010:**
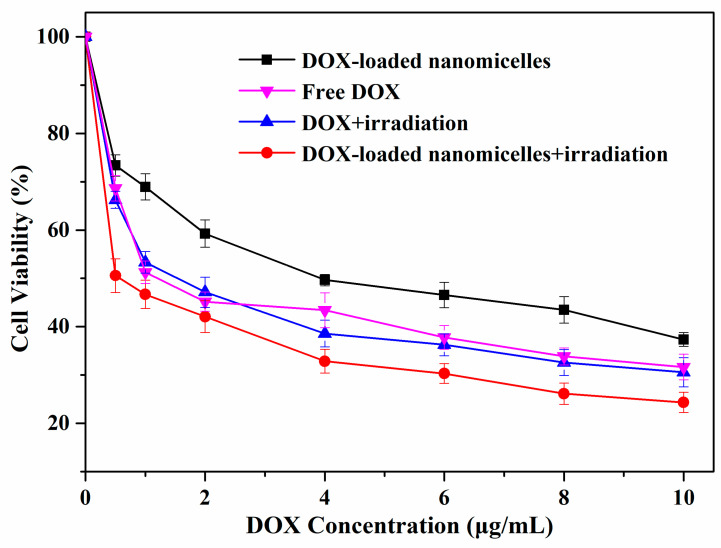
Cytotoxicity of free DOX and DOX-loaded mPEG-PEI-(NBS, OA) upconversion nanomicelles with or without irradiation at 980 nm against A549 cells.

**Figure 11 pharmaceutics-12-00680-f011:**
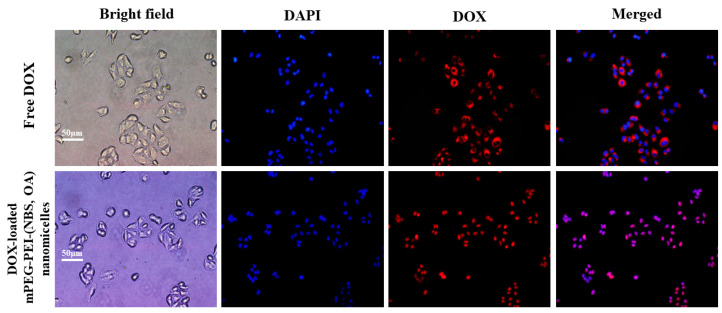
Fluorescence images of A549 cells incubated with free DOX and DOX-loaded mPEG-PEI-(NBS, OA) upconversion nanomicelles. The images from left to right show bright field (gray), cell nuclei stained by 4,6-diamidino-2-phenylindole (DAPI) (blue), DOX fluorescence (red) and overlays of the red and blue images.

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
