# Peer review of "Oleic Acid Copolymer as A Novel Upconversion Nanomaterial to Make Doxorubicin-Loaded Nanomicelles with Dual Responsiveness to pH and NIR"

_pharmaceutics, 2020, doi:10.3390/pharmaceutics12070680_

Round 1
Reviewer 1 Report
Nice work concerning drug targeting. The tumor targeting of nanomicelles responsive to pH is not enough so it can be achieved by irradiation outside the body. Under irradiation, nanomicelles are unstable and drug can be released
minor remarks:
- I am not sure about the use of the word "difficultly" in this sentence "Oleic acid (OA) as main component of plant oil is difficultly used in the nanocarrier of anticancer drugs."
- what was the precedure of "OA was distill ed from C amellia seed oil in our lab with its purity of 98%". Please explain or add reference
- the font size for the x and y axes is too large in figures 3, 6, 7, 8, 9, 10
- the list of abbreviations is missing
Author Response
Dear professors and Reviewers,
Thank you for your comments on our manuscript.
We have revised the manuscript according to your suggestions, you may track the revised parts highlighted in the manuscript. The attachment is our reply to your questions.
Best regards,
Yong Ye

Reviewer 2 Report
The article presented by Zhang et al focuses on the development of novel OA copolymer-based nanomicelles to deliver doxorubicin against A549 cells. The authors did a reasonable work in exploring the possibility of enhancing the anticancer efficacy of doxorubicin using novel nanomicelles. The experimental section was well designed with appropriate tools, and the presentation of findings was good. Overall, the article was convincing and recommended for publication after minor revision.
1. Line no 17. The sentence has to be chunked as it is too long. "The novel OA copolymer (mPEG-PEI-(NBS, OA)) was synthesized by grafting hydrophobic OA and photosensitive o-nitrobenzyl succinate (NBS) onto hydrophilic mPEGylated polyethyleneimine (mPEG-PEI) by amidation reaction, and conjugating with NaYF4: Yb3+/Er3+ nanoparticles to make the upconversion nanomicelles by self-assembly to encapsulate doxorubicin (DOX) with dual response to pH and NIR."
2. The sequence of the figure has to be modified.
Author Response

(The authors gave the same response as above.)

Reviewer 3 Report
This manuscript presents the synthesis, characterization and biological evaluation of a nanosystem based on the use of upconversion nanoparticles as core and a copolymer as shell. The shell is composed of polyethylene glycol, polyethylene imine and oleic acid. The system shows good responsiveness to both acid pH and NIR light, which can cleave a UV-responsive bond thanks to the upconversion nanoparticles. Overall, the system is biocompatible and it is able to reduce the viability of A549 cells when loaded with doxorubicin.
General comments:
The authors should emphasize the novelty of the work in the introduction.
The authors should comment on the biocompatibility of upconversion nanoparticles and whether they are easily removed from the organism.
In the experimental part, reagents are given either in grams/mL or mmol. For consistency, all should be given using both: e.g. x mg (y mmol) of…
Upconversion nanoparticles need additional characterization. For instance, thermogravimetric analysis to evaluate the amount of copolymer deposited around the particles or FTIR.
It is not clear how the final system is formed. According to Scheme 1, OA would be part of the inner core of the micelle. However, in Lines 275-277 the authors claim that the final system forms trough the coordination among the rare earth cations and the groups in NBS.
Specific points:
- Line 62: EPR does not mean that (correct: enhanced permeability and retention)
- I think the upconversion nanoparticles are missing in Scheme 1. Authors only show the micelles.
- The UV-responsiveness of NBS is a self-immolative process. Authors should include a short description of this kind of chemistry and its usefulness in biomedicine (https://doi.org/10.1039/D0TB01119C, https://doi.org/10.1016/j.cej.2017.12.098)
- Line 328-329: By saying “the mPEG copolymer approved by the FDA…”, it seems that what it is approved is the PEG-PEI copolymer, and it is not. Sentence should be “mPEG, which is approved by the FDA,…”
- Line 340: That size (and those after applying the stimuli) has been determined by DLS? If so, the graphs should be included in the manuscript.
- Explain Figure 7. Did the authors use all those concentrations and then work with the best one or…?
- Line 446: The authors claim that “The positively charged nanomicelles can be internalized by cancer cells through cell membrane because of OA’s lipophilicity and natural targeting”. Again, it is not clear to me how the final system is formed. If it is as shown in Scheme 1, OA would not be exposed to the cell membrane and, therefore, it would not participate in the internalization. The particles are positively charge, isn’t it possible that the internalization is due to electrostatic interactions?
Author Response

(The authors gave the same response as above.)
